# Adsorption of Reactive Red 120 in Decamethyl-Cyclopentasiloxane Non-Aqueous Dyeing System

**Simin Sun** [1,2], **Liujun Pei** [1,2,*] , **Jingru Chen** [1,2], **Jifang Shen** [1], **Omer Kamal Alebeid** [3], **Jianchang Xu** [4], **Chaowen Luo** [4], **Xinjie Zhang** [4], **Suqing Zhang** [4] **and Jiping Wang** [1,2,*]

[1]   Engineering Research Center of Textile Chemistry and Clean Production, Shanghai University of Engineering Science, Shanghai 201620, China
[2]   School of Textiles and Fashion, Shanghai University of Engineering Science, Shanghai 201620, China
[3]   Cleaner Production Institute, Industrial Research and Consultancy Center, Khartoum 999129, Sudan
[4]   Seduno Group Co., Ltd., Ningbo 315099, China
*   Correspondence: peilj@sues.edu.cn (L.P.); jpw@sues.edu.cn (J.W.)

**Abstract:** Traditional dyeing usually consumes a significant amount of water and salts, thus causing environmental pollution. Salt-free and low-water dyeing has become an important research direction in the cotton fabric dyeing industry. The non-aqueous media dyeing technology, using decamethylcyclopentasiloxane (D5) as the dyeing medium, has achieved energy saving and emission reduction in this industry. To investigate the influence of inorganic salts on the dyeing properties of reactive dyes in a non-aqueous medium dyeing system, the adsorption kinetics and level dyeing property of C.I. Reactive Red 120 were investigated at various concentrations of sodium sulfate. When no salts were included in the siloxane non-aqueous dyeing system, 80% of the reactive dye could diffuse onto the cotton fabric surface after 10 min. However, if 13% salts were added during dyeing, 87% of the reactive dye could diffuse to cotton fabric surface over the same amount of time. Moreover, the adsorption rate of dye was increased from 3.85 mg/g·min to 5.04 mg/g·min when the quantity of salts was increased from 0% to 13%. However, the concentration of sodium sulfate had minimal effect on the color depth of the dyed fabric and the final uptake of dye. But, when the concentration of sodium sulfate was significant, the level dyeing property of the dye became poor as the $S\gamma(\lambda)$ value was increased from 0.020 to 0.042. The adsorption kinetic of C.I. Reactive Red 120 in D5 dyeing solution may be best described by the pseudo-second-order kinetic model. As the sodium sulfate concentration increases, the half-dyeing time gradually decreases and the adsorption rate of dye increases. The repulsive force between the dye and the cotton fiber was lowered by the addition of sodium sulfate. Consequently, in the D5 dyeing system, the level dyeing property of reactive dye may be affected by the adsorption rate. Therefore, the formula of reactive dyes that do not contain salts can be applied successfully in non-aqueous dyeing systems.

**Keywords:** non-aqueous dyeing system; reactive dyes; sodium sulfate; level dyeing property; adsorption rate

## 1. Introduction

Clothing made of cotton fabric has a significant share of the textile market due to its excellent properties of biodegradability, no static electricity generation, etc. [1,2]. To colour these cotton fabrics, they are generally dyed with reactive dyes. The annual consumption of reactive dyes is about 400,000 tons [3,4]. Traditional reactive dyeing is usually performed in a water solution, as water can not only be used to fully swell fibers but also to dissolve dyes, electrolytes and other chemicals [5,6]. However, the solubility of reactive dye in water is higher, and its dye uptake is only 50%–70%. Furthermore, nearly 20%–60% of reactive dyes are hydrolyzed in traditional water based dyeing systems, leading to a lower utilization rate of the dye [6]. In order to overcome the disadvantages of the low

dyeing rate and low utilization rate of reactive dyes, it is necessary to add a significant amount of neutral electrolytes during dyeing, especially in the case of a large bath ratio [7]. Although the addition of neutral electrolytes improves the utilization rate of dyes and reduces the amount of dyes in wastewater, the presence of electrolytes as pollution in the environment becomes more serious and the dyeing wastewater is more difficult to treat, resulting in salinization of fresh water and destruction of the environment [8]. Therefore, salt-free and low-water dyeing has become an important research direction in the cotton fabric dyeing industry [9,10].

In order to realize the salt-free and low-water cleaning dyeing of cellulose fibers such as cotton, a significant amount of research has been conducted around the world [11,12]. Supercritical $CO_2$ dyeing is an eco-friendly dyeing technology that uses $CO_2$ instead of water as a dye carrier [13]. Wang and others [14–16] proposed supercritical $CO_2$ spray dyeing technology, that can realize large-volume continuous dyeing without high-temperature and high-pressure equipment, thus shortening the dyeing time and improving the dyeing efficiency. However, cotton fabric directly sprayed by the sprayer is easily damaged due to the continuous impact of the nozzle. Moreover, the sprayer has the problem of dye liquid leakage at the edge of fabric due to the poor sealing effect. As a dyeing medium, the reverse micelle system not only saves significant amounts of water, but also overcomes the need to add inorganic salt. Sawada et al. [17] studied the application of the AOT/isooctane system in reactive dyeing and found that reactive dyes could still be effectively adsorbed on the surface of cotton fibers without inorganic salts. The apparent color depth of the dyed fabric is almost the same as that of the traditional water bath dyeing. However, the preparation and storage of the reverse micelle system are also unavoidable difficulties.

He [4] and others [18,19] reported that the fabrics could obtain better dyeing properties when the alcohol volume reached an appropriate fraction in an alcohol/water dyeing system. Moreover, this dyeing system can be operated at a low temperature with less salt and less water. The dye is more effectively utilized since the uptake is nearly 100% and the fixation rate is higher than with the traditional water. However, alcohol is volatile and flammable, meaning it cannot be widely used in textile dyeing factories. Shao [20] found that the liquid paraffin has better stability, was significant safer and had no toxic effect on the human body. Moreover, high fixation and dye uptake can be obtained with this dyeing system. However, the viscosity of liquid paraffin is relatively high, making it difficult to completely wash from dyed fabrics [21]. Therefore, more water is consumed during the washing process.

To achieve environmentally friendly dyeing technology with less water consumption, a new non-aqueous dyeing medium using decamethyl cyclopentasiloxane was investigated [22–24]. As a non-aqueous medium, siloxane is non-toxic, tasteless and easy to recycle [25,26]. In a siloxane non-aqueous dyeing system, the diffusion of dyes involves several processes. Firstl reactive dye is dissolved in a small amount of water. Due to the high affinity between dye solution and cotton fabric, the high concentration of dye solution dispersed in the siloxane dyeing system diffuses to the fiber [27,28]. A water bath dyeing environment with a small bath ratio is formed on fabric surface. Reactive dye will diffuse into the fiber due to the concentration gradient and then bond with the fiber under alkaline conditions [29,30]. Compared with a water dyeing bath, the final exhaustion of reactive dye more than doubled in the D5 dyeing system. The fixation dye was almost above 90% and the color depth of dye was greatly improved [22,31]. Additionally, because the viscosity of siloxane dyeing is low, it is easy to wash away after dyeing. Since siloxane is incompatible with water, it can be reused after a simple static separation [32,33]. Therefore, the siloxane non-aqueous dyeing system has great industrial application prospects. However, to obtain good level dyeing property, commercial reactive dye must be purified and then used in the D5 dyeing system. The reason maybe that the inorganic salts of the commercial dye formulation influence the diffusion kinetic of dye, which in turn affects the level dyeing of dye.

In this paper, environmentally friendly D5 as the dyeing medium was used instead of the traditional water medium. To explore the influence of inorganic salts on the dyeing

performance of dye and investigate the adsorption of reactive dye in a silicone non-aqueous medium dyeing system, the defects of KE-type reactive dyeing caused by inorganic salts were further explained by dyeing kinetic adsorption in the D5 dyeing system. Moreover, the influence of inorganic salts on the dyeing level property of dye and the dyeing affinity between cotton fiber and dye were investigated. These successful investigations of the influence of inorganic salts on the dyeing properties of reactive dyes in non-aqueous medium dyeing systems can significantly aid development of reactive dye formulations for non-aqueous medium dyeing technology.

## 2. Experimental

### 2.1. Materials

Woven plain cotton (gram weight: 127.2 g/m$^2$, yarn count: 130 D × 130 D, warp and weft density: 287 × 146 root/10 cm), D5 and reactive dyes were obtained from Zhejiang Green Universe Textile Technology Co., Ltd. (Hangzhou, China). The molecular structure of the dye is shown in Figure 1. The reactive dye, that had been further purified before the dyeing process, had a maximum absorbance peak at a wavelength of 536 nm. To maintain the pH at 10–11 during the dyeing process, sodium carbonate (Analytical Reagent) was acquired from Hangzhou Gaojing Fine Chemical Co., Ltd. (Hangzhou, China). Standard soap tablets (Mien Tester Co., Ltd., Shanghai, China) was used to wash the dyed fabric. Anhydrous sodium sulfate (Analytical Reagent) was purchased from Shanghai Titan Co., Ltd. (Shanghai, China).

**Figure 1.** Molecule structure of C.I. Reactive Red 120.

### 2.2. Reactive Dyeing Process

Reactive dyeing in the D5 dyeing system was performed in a DYE-24 dyeing machine (ShangHai Chain-Lih Automation Equipment Co., Ltd., Shanghai, China). The fabric sample weight was 2.0 g. The dosage of dye was 2.0% o.w.f (on the weight of fabric) and the sodium carbonate concentration was 30 g/L. The dosage of water was 130% (o.w.f). The dosage of sodium sulfate was 2.6%, 5.2%, 7.8%, 10.4% and 13%, respectively (o.w.f). The bath ratio of the D5 to fabric was 30:1. Dyeing was started at 25 °C and maintained for 10 min. Then, the temperature was raised to 90 °C at a rate of 2 °C/min. After 30 min, the dyed sample was washed with 2.0 g/L of standard soap flakes in a soap/water solution at 95 °C for 15 min, then washed thoroughly for 15 min in a constant temperature oscillating water bath at room temperature. Finally, the dyed samples were dried at 60 °C for 2 h.

### 2.3. Determination of K/S Value of Dyed Cotton Fabric

The Kubelka–Munk function, Equation (1), was used for calculating the K/S value of dyed fabrics.

$$K/S = (1 - R)^2/2R \tag{1}$$

where *K* represents the absorption coefficient of the measured object and *S* represents the scattering coefficient of the measured sample [34]. *R* is the wavelength of minimum reflectance (maximum absorbance). This was tested on a Datacolor SF600X spectrophotometer (Datacolor, Lawrenceville, NJ, USA). Each fabric sample was tested 5 times on different parts and the average value was employed for the dyeing color depth.

### 2.4. Level Dyeing Property

Twelve points were randomly selected to test their K/S values by using a Datacolor SF600X spectrophotometer. The level dyeing property was tested by Equations (2) and (3).

$$S_\gamma(\lambda) = \sqrt{\frac{\sum\limits_{i=1}^{n} \frac{(K/S)_{i\lambda}}{(\overline{K/S})_{i\lambda}} - 1}{n-1}} \tag{2}$$

$$\overline{(K/S)}_\lambda = \frac{1}{n}\sum_{i=1}^{n}(K/S)_{i\lambda} \tag{3}$$

where $S\gamma(\lambda)$ denotes the level dyeing property of dyed cotton fabric. The higher the value, the worse the level dyeing property.

### 2.5. The Calibration Curve of Reactive Red 120

A measurement of 0.20 g of reactive red 120 was dissolved in deionized water and determined to be 500 mL; this was the primary solution. Then 1, 3, 6, 7 and 9 mL of the primary solution was measured and placed into a 50 mL volumetric bottle. The absorption wavelength of the dye solution was measured with a UV-visible spectrophotometer (UV-2600 Shimadzu Enterprise Management Co., Ltd., Shanghai, China). The calibration curve of dye was established by taking the mass concentration of dye as the abscissa and the absorbance of dye at the maximum absorption wavelength as the ordinate.

### 2.6. Determination of the Final Uptake of Dye

The final uptake of dye refers to the percentage of dye adsorbed on the fabric in the initial dyeing bath during dyeing. The residual liquid method was employed to calculate the percentage of dyeing. The concentration of dye was determined with a UV-visible spectrophotometer (UV-2600 Shimadzu Enterprise Management Co., Ltd. Suzhou, China). Samples of dye solutions were taken at different time (1, 3, 5, 7, 10, 20, 40, 50, 60 and 70 min) for UV measurements of the maximum absorption wavelength of dye. The expression of final uptake of dye is shown in Equation (4) [35].

$$E = \left(1 - \frac{A_1}{A_0}\right) \times 100\% \tag{4}$$

where $E$ refers the final uptake of dye, $A_0$ refers the absorbance of the freshly prepared dyeing solution and $A_1$ refers the absorbance of the dyeing residue at a certain time.

### 2.7. Dyeing Kinetics

The reactive dye was maintained in the D5 dyeing system for 70 min at 30 °C to research the dyeing kinetics. The data conformity was assessed through the investigation of the pseudo-first-order and pseudo-second-order models.

The pseudo-first-order model assumes that the rate change of dye exhaustion with time is correlated with the difference in saturation concentrations [36,37]. The pseudo-second-order model assumes that the adsorption rate determining step may be a chemical surface reaction [38–40].

The pseudo-first-order model follows Equation (5).

$$\frac{dq_e}{dt} = k_1(q_e - q_t) \tag{5}$$

The pseudo-second-order model follows Equation (6) [41–43].

$$\frac{dq_e}{dt} = k_2(q_e - q_t)^2 \tag{6}$$

where $q_e$ (mg/g) and $q_t$ (mg/g) represent the masses of dye that was adsorbed on cotton sample at equilibrium time and at a certain time $t$ (min), respectively, and $k_1$ (min$^{-1}$) is the rate constant of the pseudo-first-order kinetic model. $k_2$ (g/mg·min$^{-1}$) is rate constant of the pseudo-second-order kinetic model.

The half-dyeing time ($t_{0.5}$), which is the length of time needed for half of the dye to reach equilibrium adsorption, can also be used to represent the dyeing rate. The $t_{0.5}$ was calculated using Equation (7).

$$t_{0.5} = \frac{1}{k_2 \cdot q_e} \tag{7}$$

### 2.8. Color Fastness

The samples that had been dyed were evaluated for colorfastness to washing and rubbing using ISO 105-C06 and ISO 105-E04:2013 testing standards, respectively. Colorfastness to washing was assessed in respect of color change in the sample and staining on the multifiber fabric. Rubbing fastness was evaluated in dry and wet conditions.

## 3. Results and Discussion

### 3.1. Effect of Salt Concentration on the Dye Diffusion

Figure 2 shows the diffusion rate of C.I. Reactive Red 120 with different concentration of salts in the siloxane non-aqueous dyeing system. After a dyeing time of 5 min, 70% of the dye was adsorbed on the fabric, indicating that the amount of dye on the fabric surface gradually increased per unit time. After 20 min, more than 95% of the dye completed the adsorption process, indicating that almost all of the dye reached adsorption equilibrium. Therefore, the concentration of sodium sulfate can increase the adsorption rate of C.I. Reactive Red 120 in the siloxane non-aqueous dyeing system. C.I. Reactive Red 120 can achieve almost 100% uptake of dye with or without salt. This interesting phenomenon is like because the sodium sulfate weakens the charge repulsion between the dye and the fiber, further influencing the diffusion rate of dye [44]. The final uptake had few changes in siloxane non-aqueous dyeing system, because it was dependent on the amount of water.

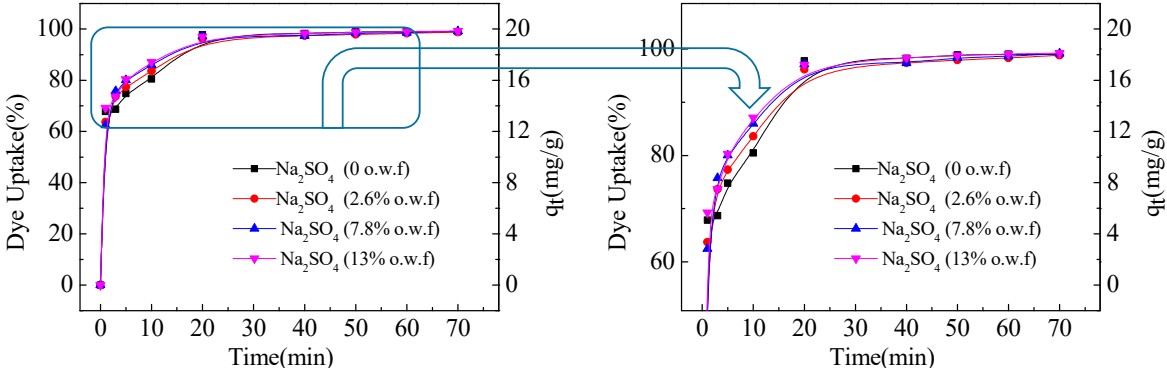

**Figure 2.** Uptake of dye of C.I. Reactive Red 120 with different sodium sulfate concentrations.

### 3.2. Influence of Salt Concentrations on the Dyeing Properties of Dye

Figure 3 showed the K/S value and the level dyeing property (Sγ(λ)) of dyed fabrics with different concentrations of sodium sulfate in the D5 dyeing system. As the salt concentration in the C.I. Reactive Red 120 formulation increased, the K/S value of dyed fabric showed a slight increase. For example, the K/S value of dyed fabric was 15.89 when there was no salt during dyeing. When the amount of salts was increased to 5.2% (o.w.f), the K/S value of the dyed sample was 16.67. That suggests that, in this dyeing system, the amount of salt in the dyeing formula has little influence on the K/S value of the dyed fabric. The levelness of the dyed fabric was shown in Figure 3b. When there was no salt in C.I. Reactive Red 120 formulation, the Sγ(λ) value of the dyed sample was 0.02. However, the Sγ(λ) value gradually increased alongside the increase of the salt concentration. As a result,

the level dyeing property of the dyed fabric gradually decreased when the concentration of salt increased. The reason for this may be that, with the increase of salt concentration in the dye formulation, the diffusion rate of dye becomes faster, resulting a worse level dyeing property of the dyed sample. Therefore, considering the K/S value and $S\gamma(\lambda)$ of dyed fabrics, dyeing using no salts can be achieved in a siloxane non-aqueous dyeing system.

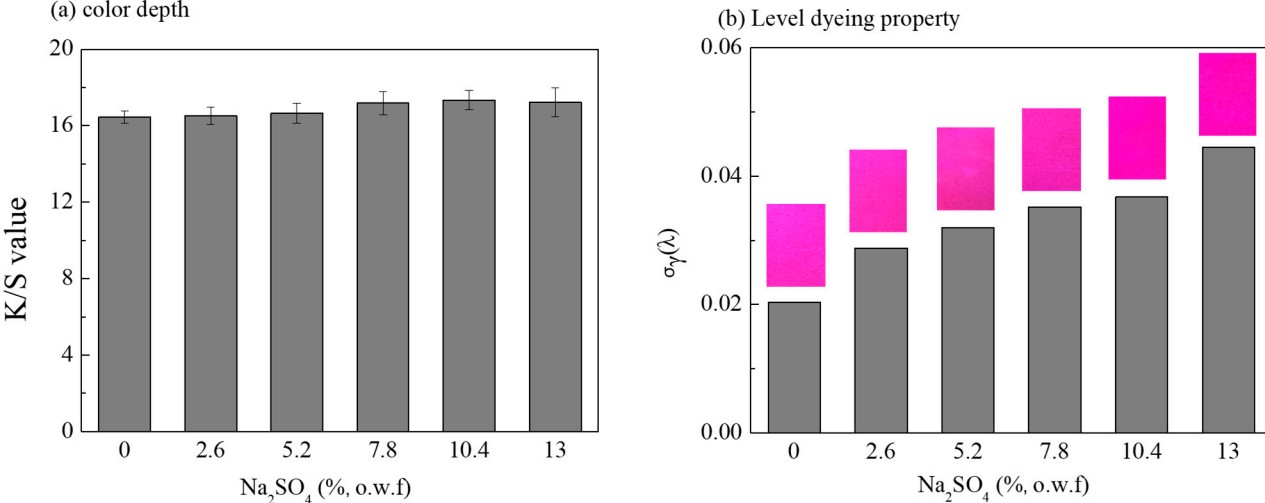

**Figure 3.** K/S value (**a**) and level dyeing property (**b**) of dyed fabrics under different $Na_2SO_4$ concentrations in siloxane non-aqueous dyeing system.

Cotton fabric was dyed with Reactive Red 120 with different concentrations of sodium sulfate under the same conditions. The results comparing the effects of various sodium sulfate concentrations on the color fastness of dyed fabrics are displayed in Table 1. The color fastness of dyed cotton fabric subjected to dry rubbing was higher than 4 and the wet rubbing was 3~4 or 4. Color fastness to cotton and polyester was higher than 4, indicating that all kinds of color fastness were excellent under different salt concentrations. As a result, in a non-aqueous medium dyeing technique, a varied salt concentration has no effect on the fastness qualities of cotton fabric. The product quality is completely on par with international standards.

**Table 1.** Color fastness of dyed fabric.

| Dye | $Na_2SO_4$ Concentration (o.w.f) | Rubbing | | Washing | | |
|---|---|---|---|---|---|---|
| | | Dry | Wet | Color Change | Cotton | Polyester |
| Red 120 | 0 | 4 | 3~4 | 4~5 | 4~5 | 4~5 |
| | 2.6 | 4 | 3~4 | 4~5 | 4~5 | 4~5 |
| | 5.2 | 4 | 3~4 | 4~5 | 4~5 | 4~5 |
| | 7.8 | 4~5 | 4 | 4~5 | 4~5 | 4~5 |
| | 13.0 | 4~5 | 4 | 4~5 | 4~5 | 4~5 |

### 3.3. Dyeing Adsorption Kinetics

#### 3.3.1. Fitting of Pseudo-First-Order Kinetic Model

The pseudo-first-order kinetics fitting line plots and coefficient $R^2$ of cotton fabrics dyed with different concentrations of sodium sulfate in non-aqueous medium dyeing are shown in Figure 4 and Table 1. The pseudo-first-order kinetic model cannot reliably predict the impact of sodium sulfate concentration on the adsorption of reactive dye, as shown by the fitting coefficients $R^2$ (Table 2), that were all less than 0.99.

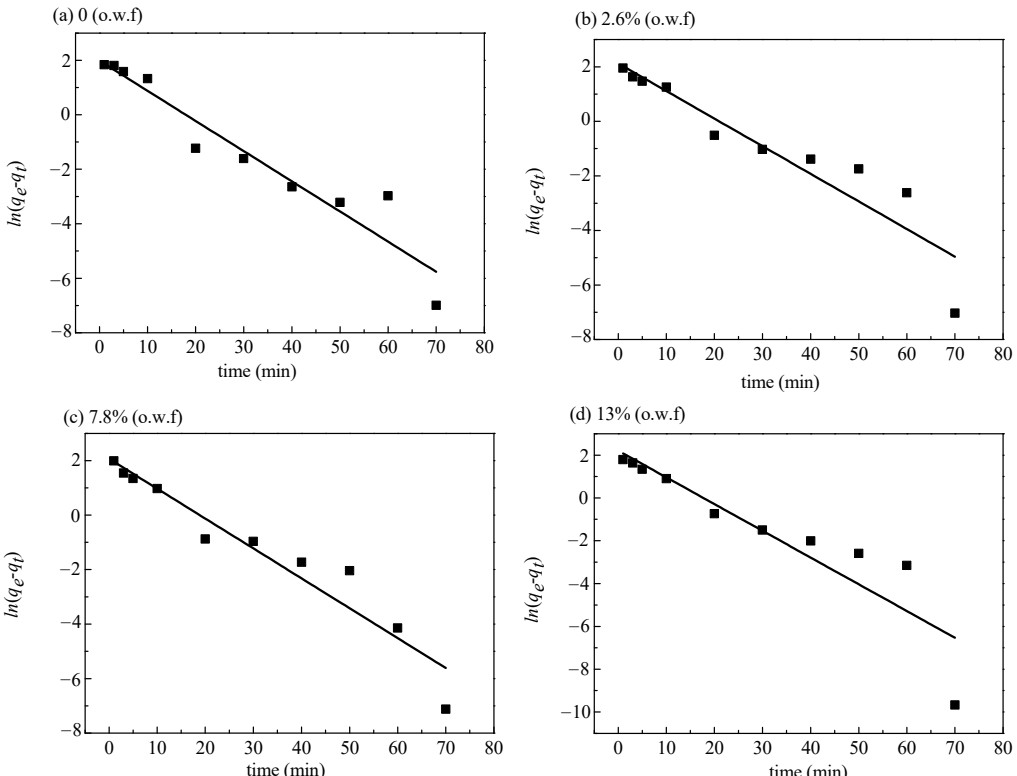

**Figure 4.** Pseudo-first-order kinetic fitting curve of reactive dyeing with different concentrations of sodium sulfate ((**a**) 0%, (**b**) 2.6%, (**c**) 7.8%, (**d**) 13%) in siloxane non-aqueous dyeing system.

**Table 2.** $R^2$ of pseudo-first-order kinetic model of reactive dyeing with different sodium sulfate concentration.

| Na$_2$SO$_4$ Concentration (o.w.f) | 0 | 2.6% | 7.8% | 13% |
|---|---|---|---|---|
| R$^2$ | 0.9133 | 0.8604 | 0.9183 | 0.8147 |

3.3.2. Fitting of Pseudo-Second-Order Kinetic Model

The fitting curves about $t/q_t$ versus t were shown in Figure 5 and kinetic parameter was shown in Table 3. The pseudo-second-order kinetic fitting coefficients ($R^2$) were all greater than 0.999 under different concentration of sodium sulfate. Moreover, the experimental adsorption equilibrium data ($q_{e,exp}$) fitted well with the theoretical adsorption capacities ($q_{e,cal}$), indicating that the adsorption of C.I. Reactive Red 120 could be well evaluated by pseudo-second-order dynamic model. The amount of dye adsorbed on the fabric surface was close to the dye adsorption equilibrium value by theory (20 mg/g). It further indicated that the ultimate adsorption of dye was not greatly affected by sodium sulfate concentration.

Compared with the pseudo-first-order kinetic model, the the fitting coefficients ($R^2$) of pseudo-second-order kinetic model is suitable for describing the adsorption of dye in D5 dyeing system. Therefore, the adsorption rate of high concentration dye solution on cotton fabric is not proportional to the dye concentration, but it is determined by the square value of the numbers of vacant adsorption spots on the fiber surface [45,46]. It is well known that reactive dye can bind to cotton fiber in the form of covalent bonds. Therefore, reactive dye can diffuse to the inner of fiber, and then react with cotton fiber under an alkaline environment. All the reactions are occurred the amorphous region of the cotton fiber.

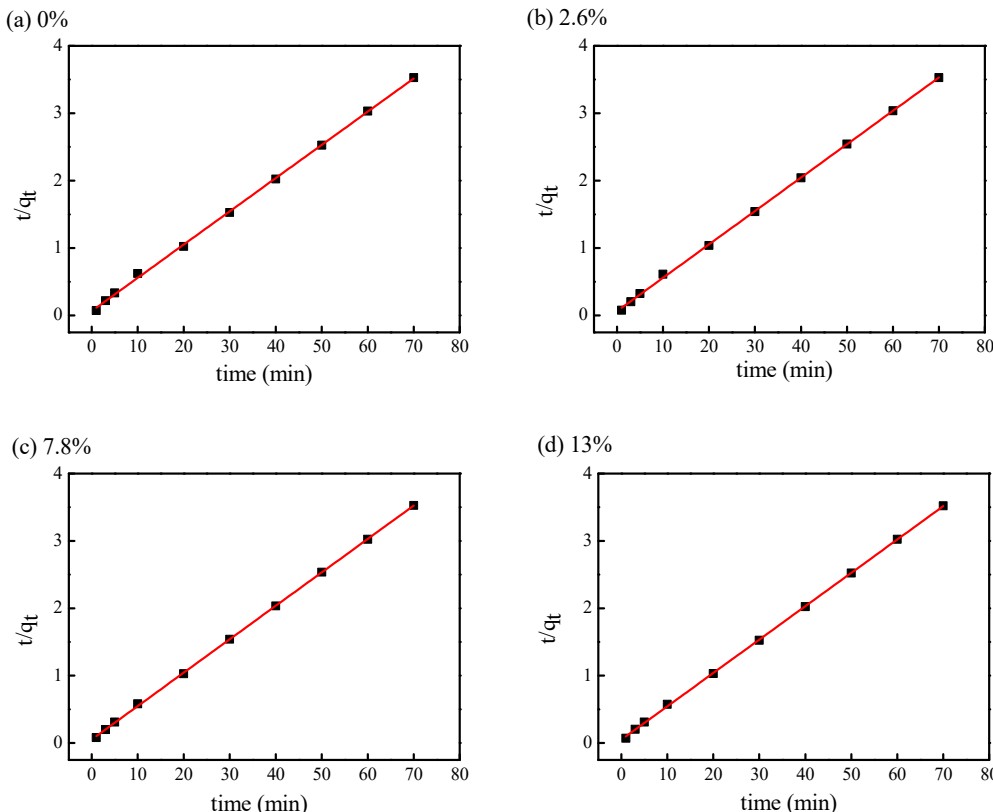

**Figure 5.** Pseudo-second-order kinetic fitting curve with different concentrations of sodium sulfate ((**a**) 0%, (**b**) 2.6%, (**c**) 7.8%, (**d**) 13%) in D5 dyeing system.

**Table 3.** Kinetic parameters of pseudo-second-order for adsorption of C.I. Reactive Red 120 with different concentrations of sodium sulfate in D5 dyeing system.

| Na$_2$SO$_4$ Concentration | 0 | 2.6% | 7.8% | 13% |
|---|---|---|---|---|
| $K_2 \times 10^{-2}$ (mg/g·min) | 3.8500 | 4.0036 | 4.7796 | 5.0419 |
| $q_{e,exp}$ (mg/g) | 19.8440 | 19.8330 | 19.8480 | 19.8870 |
| $q_{e,cal}$ (mg/g) | 20.2881 | 20.1857 | 20.1654 | 20.1979 |
| $t_{0.5}$ (min) | 1.3089 | 1.2594 | 1.0541 | 0.9973 |
| $R^2$ | 0.9994 | 0.9996 | 0.9998 | 0.9998 |

As the concentration of sodium sulfate increased, the dyeing rate constant ($K_2$) gradually increased. The dye adsorption rate was 3.8500 × 10$^{-2}$ g/(mg·min) when sodium sulfate was absent from the dye formulation. However, when the concentration of sodium sulfate was 13% (o.w.f), the dye adsorption rate was increased to 5.0419 × 10$^{-2}$ g/(mg·min) 1.31 times the rate without sodium sulfate. Meanwhile, the half-dyeing time $t_{0.5}$ decreased with the increased concentration of sodium sulfate. As a result, the higher the concentration of sodium sulfate in the D5 dyeing system, the faster the dye adsorption rate and the shorter time to reach equilibrium. The faster dyeing adsorption rate might influence the level dyeing property of dyed fabric.

### 3.4. Dyeing Thermodynamic of Reactive Red 120 in Non-Aqueous Medium Dyeing System

Dyeing affinity ($-\Delta\mu^\circ$) is the tendency of a dye to transfer from a dyeing bath to a fiber surface under a standard dyeing solution. The greater the affinity, the greater the transfer trend of dye from dye solution to fiber, that is, the greater the driving force [47]. Driving force can be calculated using Equation (8).

$$-\Delta\mu^\circ = RT \ln C_f / C_s \tag{8}$$

where $C_f$ and $C_s$ denote the dye concentration at fiber and dyeing bath, respectively. If the changes of affinity were calculated, the heat of dyeing ($\Delta H^\circ$) and entropy ($\Delta S^\circ$) can be calculated according to Equation (9).

$$-\Delta\mu^\circ = T\Delta S^\circ - \Delta H^\circ \tag{9}$$

When the dyeing temperature was not changed, the value of $\Delta H^\circ$ is a constant and the relationship between $-\Delta\mu^\circ$ was plotted temperature T to obtain a straight line. The slope and intercept of the line can be used to determine the value of $\Delta S^\circ$ and $\Delta H^\circ$.

As shown in Table 4, the dyeing affinity between cotton fiber and dye was 5.85 KJ/mol at 20 °C when there were no salts in the D5 dyeing system. As the temperature rises, the dyeing affinity between cotton fiber and dye was increased. For example, the dyeing affinity between cotton fiber was changed from 5.85 KJ/mol to 11.10 KJ/mol when the dyeing temperature was raised from 20 °C to 60 °C. Compared with different concentrations of salts, the dyeing affinity between cotton fiber and dye was increased from 5.85 KJ/mol to 9.65 and 11.62 and 12.67 KJ/mol when the concentration of salt was increased from 0 to 5.2% and 7.8% and 13%, respectively. All the values $\Delta H^\circ$ of dyeing were negative, indicating that dyeing using reactive dyes in a non-aqueous medium dyeing system is an exothermic reaction. Moreover, the amount of heat released increased with the amount of salt. The value of $\Delta S^\circ$ was also negative, but it was low, indicating that the motion of dyes was low in the non-aqueous medium dyeing system. Therefore, the repulsive attraction between the dye and the fiber is weakened by the addition of sodium sulfate. This repulsive force continues to weaken as sodium sulfate concentration rises, accelerating dye diffusion.

**Table 4.** Thermodynamic parameters of dye diffusion under different salts concentration.

| Salt Concentration (o.w.f) | Temperature (°C) | $\Delta\mu^\circ$ (KJ/mol) | $\Delta H^\circ$ (KJ/mol) | $\Delta S^\circ$ (KJ/K·mol$^{-1}$) |
|---|---|---|---|---|
| 0 | 20 | 5.85 | −25.28 | −0.11 |
|  | 40 | 8.03 |  |  |
|  | 60 | 10.10 |  |  |
| 5.2% | 20 | 9.65 | −27.62 | −0.13 |
|  | 40 | 11.20 |  |  |
|  | 60 | 14.69 |  |  |
| 7.8% | 20 | 11.62 | −30.20 | −0.14 |
|  | 40 | 13.05 |  |  |
|  | 60 | 17.26 |  |  |
| 13% | 20 | 12.67 | −31.81 | −0.15 |
|  | 40 | 14.31 |  |  |
|  | 60 | 18.68 |  |  |

## 4. Conclusions

This study investigates the adsorption equilibrium, kinetics and level dyeing property of C.I. Reactive Red 120 on cotton fabrics with different concentration of salts. In the D5 dyeing system, reactive dye can quickly adsorb on cotton fabric surface. After 20 min, more than 95% of the dye completed the dyeing process. Compared with salts-free, salt has little effect on the K/S value of the dyed fabric and the final uptake of dye was close to 100% where there was a small amount of salt or no salt at all in the dyeing formulation. Moreover, the color fastness of dyed cotton fabric was at a level of 3~4 or 4. However, salts can influence the adsorption rate of reactive dye. When there were no salts in the siloxane non-aqueous dyeing system, the dye adsorption rate was $3.85 \times 10^{-2}$ g/(mg·min). However, when the concentration of salt was increased to 13% (o.w.f), the dye adsorption rate increased to $5.04 \times 10^{-2}$ g/(mg·min). The half-dyeing time ($t_{0.5}$) decreased from 1.31 min to 1.00 min when the concentration of sodium sulfate was increased from 0% to 13%. Therefore, the adsorption rate of dye was increased if salts were used in the siloxane non-aqueous medium dyeing system.

Because the Sγ(λ) value increased from 0.020 to 0.042 when 13% salts were used during dyeing, the level dyeing property of dyed fabric dyed using reactive dye was gradually decreased as the salt concentration increased. Therefore, the adsorption rate of dye was accelerated, leading to a short dyeing adsorption time. The fast adsorption rate may influence the level dyeing property of reactive dye. In conclusion, a better dyeing performance can be achieved without salt in the D5 dyeing system.

**Author Contributions:** Methodology, S.S., L.P., J.C. and J.W.; investigation, J.S.; resources, L.P. and J.W.; writing—original draft preparation, J.C.; writing—review and editing, L.P., O.K.A., J.X., C.L., X.Z., S.Z. and J.W. All authors have read and agreed to the published version of the manuscript.

**Funding:** This work was supported by the National Natural Science Foundation of China (22072089) and Key Research and Development Program of Xinjiang Production and Construction Corps (2019AA001).

**Institutional Review Board Statement:** Not applicable.

**Informed Consent Statement:** Not applicable.

**Data Availability Statement:** The data presented in this study are available on request from the corresponding author.

**Conflicts of Interest:** The authors declare no conflict of interest.

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
