# Peer review of "Adsorption of Reactive Red 120 in Decamethyl-Cyclopentasiloxane Non-Aqueous Dyeing System"

_coatings, doi:10.3390/coatings13030502_

Round 1

Reviewer 1 Report

The manuscript is about an important topic of addressing the wastewater issue in dyeing, by proposing alternative solutions like with decamethyl cyclopentasiloxane non-aqueous medium. Overall, it is a well-written draft, though detailed proofreading and addressing the following comments are necessary.

For example, line 15, should be... various concentrations...

line 20, should be.. With the increase of the sodium sulfate

line 22, should be... may influence...

line 41, should be ... electrolytes improve..

line 48, should be ... a lot of research...

line 63, should be... an appropriate...

and more throughout the draft. Therefore, detailed proofreading is necessary.

Moreover,

  1. There is a lack of discussion why second-order kinetic was a better fit than the first-order. This should be discussed in relation to the chemical interaction during adsorption. 
  1. To justify that second order was a really good fit, at least one of the other types of second-order equations should be verified (there are 4 types). This is because the second-order kinetics sometimes show a very good fit only because of the presence of time on both sides.  The following papers could be helpful, DOI: 10.15376/biores.14.3.7582-7626, https://doi.org/10.3390/su141711098. There is no need to present the new fitting in the manuscript, but it can be done in background calculations and then can be mentioned in the text as a note.  
  1. Please include intraparticle diffusion analysis or mention why it was not necessary.

Reviewer 2 Report

The work presents the possibility of using a new dyeing medium based on decamethyl cyclopentasiloxane that can replace the aqueous medium.

The manuscript is well written, but the following aspects need to be clarified:

1. The novelty of the work must be specified in the Introduction section.

2. In the materials and methods section, a paragraph specifying the methods of analysis and how the calibration curve was obtained should be included.

3. The conclusions must be different from the abstract.

4. In the abstract, the obtained results should be briefly specified.

5. In the materials and methods section, a paragraph specifying the methods of analysis and how the calibration curve was obtained should be included.

Author Response

The work presents the possibility of using a new dyeing medium based on decamethyl cyclopentasiloxane that can replace the aqueous medium.

The manuscript is well written, but the following aspects need to be clarified:

Comment 1: The novelty of the work must be specified in the Introduction section.

Response : We appreciate the reviewer’s comment.The novelty of the work has been added in the Introduction.

Change:

To explore the influence of inorganic salts on the dyeing performance of dye and investigate the adsorption of reactive dye in silicone non-aqueous medium dyeing system, the defects ...

These successful investigations of the influence of inorganic salts on the dyeing properties of reactive dyes in non-aqueous medium dyeing system can greatly develop reactive dye formulation for non-aqueous medium dyeing technology.

Comment 2: In the materials and methods section, a paragraph specifying the methods of analysis and how the calibration curve was obtained should be included.

Response : We appreciate the reviewer’s comment.The calibration curve of reactive dye has been added in the Introduction.

Change:

2.5. The calibration curve of reactive red 120

0.20 g of reactive red 120 was dissolved in deionized water and determined to 500 mL, which was the primary solution. Then 1, 3, 6, 7 and 9 mL of primary solution was measured and taken into 50 mL of volumetric bottle. The absorption wavelength of dye solution was measured with a UV-visible spectrophotometer (UV -2600 Shimadzu Enterprise Management Co., Ltd.). The calibration curve of dye was established by taking the mass concentration of dye as the  abscissa and the absorbance of dye at the maximum absorption wavelength as the ordinate.

Comment 3: The conclusions must be different from the abstract.

Response : We appreciate the reviewer’s comment. The conclusions was improved in manuscript.

Change:

This study investigates the adsorption equilibrium, kinetics and level dyeing property of C.I. Reactive Red 120 on cotton fabrics with different concentration of salts. In siloxane non-aqueous dyeing system, reactive dye can quickly diffuse to the surface of cotton fabric. After 20 min, above 95% of dye completed the dyeing process. Compared with salts-free, the concentration of sodium sulfate has little effect on the color depth of the dyed fabric, and the final uptake of dye is close to 100% no matter there is a small amount of salt or no salt in the dyeing formulation. Moreover, the color fastness of dyed cotton fabric was at level of 3~4 or 4. However, salts can influence the adsorption rate of reactive dye. When there was no salts in siloxane non-aqueous dyeing system, the dye adsorption rate was 3.85×10-2 g/(mg·min). However, when the concentration of sodium sulfate was 13% (o.w.f), the dye adsorption rate was increased to 5.04×10-2 g/(mg·min), which was 1.31 times that without sodium sulfate. The half-dyeing time (t0.5) was decreased from 1.31 min to 1.00 min when the concentration of sodium sulfate was increased from 0% to 13%. Therefore, the adsorption rate of dye was increased if salts was used in siloxane non-aqueous medium dyeing system. Because the Sγ(λ) value was increased from 0.020 to 0.042 when 13% of salts was used during dyeing, the level dyeing property of dyed fabric of reactive dye was gradually decreased with the increase of salt concentration. Therefore, the adsorption rate of dye was accelerated, which leaded to a short dyeing adsorption time. The fast adsorption rate may influence the level dyeing property of reactive dye. In conclusion, a better dyeing property can be obtained without salt in siloxane non-aqueous dyeing system.

Comment 4: In the abstract, the obtained results should be briefly specified.

Response : We appreciate the reviewer’s comment.The abstract was improved in manuscript.

Change:  

Traditional dyeing usually consumes a large amount of water and salts, thus causing environmental pollution. Therefore, salt-free and low-water dyeing has become an important research direction in the cotton fabric dyeing industry. The non-aqueous media dyeing technology using decamethylcyclopentasiloxane as the dyeing medium has achieved energy saving and emission reduction dyeing for cotton textiles. To investigate the influence of inorganic salts on the dyeing properties of reactive dyes in non-aqueous medium dyeing system, the adsorption kinetics and level dyeing property of C.I. Reactive Red 120 were investigated at various concentrations of sodium sulfate. When there was no salts in siloxane non-aqueous dyeing system, 80% of reactive dye could diffuse to cotton fabric surface after 10 min. However, if 13% of salts was added during dyeing, 87% of reactive dye could diffuse to cotton fabric surface at the same time. Moreover, the adsorption rate of dye was increased from 3.85 ×10-2 mg/g·min to 5.04 ×10-2 mg/g·min when the amount of salts was increased from 0% to 13%. However, ...

Comment 5: In the materials and methods section, a paragraph specifying the methods of analysis and how the calibration curve was obtained should be included.

Response : We appreciate the reviewer’s comment.The calibration curve of reactive dye has been added in the Introduction.

Change:

2.5. The calibration curve of reactive red 120

0.20 g of reactive red 120 was dissolved in deionized water and determined to 500 mL, which was the primary solution. Then 1, 3, 6, 7 and 9 mL of primary solution was measured and taken into 50 mL of volumetric bottle. The absorption wavelength of dye solution was measured with a UV-visible spectrophotometer (UV -2600 Shimadzu Enterprise Management Co., Ltd.). The calibration curve of dye was established by taking the mass concentration of dye as the  abscissa and the absorbance of dye at the maximum absorption wavelength as the ordinate.

Reviewer 3 Report

This manuscript is a good addition to the literature. However, it requires minor revisions as follows:

1. The abstract should be refined and rewritten as there are few grammatical errors.

2. As the authors claim salt free dyeing, but from experimental results the rate of dye uptake increases with increase in sodium sulfate concentration.

3. Figure 2: for comparison the axis reading should be same.

Reviewer 4 Report

Dear Editor and Authors,

The study is interesting and I recommend the publication of the paper in the present format. 

Author Response

Response : Thank you so much for your kind comments and encouragement. 

Reviewer 5 Report

The paper investigates the influence of sodium sulfate on the dyeing properties of Reactive Red 120 dye in a siloxane solvent (and water ?) on fabrics.

The paper is well written, but the link between the dyeing process and dyed fabric results is unclear. In particular:

11. The procedure described in Section 2.2. “Reactive Dyeing Process” is unclear. As well as in the whole manuscript it is not clear when water is used or not.

22. The title “Salts-free and Adsorption of Reactive Red 120 in Decamethyl-2 cyclopentasiloxane Non-aqueous Dyeing System” is misleading because a salt was used, as well as water.

33.  Abstract and Conclusions must be rewritten to clarify the results obtained.

44. The Authors carried out a comparison between fabrics dyed with water and fabrics dyed in the non-aqueous system.

55.  I suggest labeling the samples.

Round 2

Reviewer 2 Report

From my point of view the manuscript can be accepted for publication.